# Diverse Secondary Metabolites from the Coral-Derived Fungus *Aspergillus hiratsukae* SCSIO 5Bn_1_003

**DOI:** 10.3390/md20020150

**Published:** 2022-02-18

**Authors:** Qi Zeng, Yuchan Chen, Junfeng Wang, Xuefeng Shi, Yihao Che, Xiayu Chen, Weimao Zhong, Weimin Zhang, Xiaoyi Wei, Fazuo Wang, Si Zhang

**Affiliations:** 1CAS Key Laboratory of Tropical Marine Bio-Resources and Ecology, Southern Marine Science and Engineering Guangdong Laboratory (Guangzhou), Guangdong Key Laboratory of Marine Materia Medica, RNAM Center for Marine Microbiology, South China Sea Institute of Oceanology, Chinese Academy of Sciences, 164 West Xingang Road, Guangzhou 510301, China; zengqi15@mails.ucas.ac.cn (Q.Z.); wangjunfeng@scsio.ac.cn (J.W.); shixuefeng@scsio.ac.cn (X.S.); cheyihao1995@163.com (Y.C.); chenxiayu17@mails.ucas.ac.cn (X.C.); wmzhong@scsio.ac.cn (W.Z.); 2University of Chinese Academy of Sciences, 19 Yuquan Road, Beijing 100049, China; 3State Key Laboratory of Applied Microbiology Southern China, Guangdong Provincial Key Laboratory of Microbial Culture Collection and Application, Institute of Microbiology, Guangdong Academy of Sciences, 100 Central Xianlie Road, Guangzhou 510070, China; chenyc@gdim.cn (Y.C.); wmzhang@gdim.cn (W.Z.); 4Key Laboratory of Plant Resources Conservation and Sustainable Utilization, South China Botanical Garden, Chinese Academy of Sciences, Guangzhou 510650, China; wxy@scbg.ac.cn

**Keywords:** coral-derived fungi, *Aspergillus hiratsukae*, structure elucidation, bioactivity evaluation

## Abstract

Three new metabolites, including a cyclic tetrapeptide asperhiratide (**1**), an ecdysteroid derivative asperhiratine (**2**), and a sesquiterpene lactone asperhiratone (**3**), were isolated and identified from the soft coral-derived fungus *Aspergillus hiratsukae* SCSIO 5Bn_1_003, together with 10 known compounds. Their structures were elucidated via spectroscopic analysis, X-ray diffraction analysis, and electronic circular dichroism calculations. In addition, the absolute configuration of **1** was determined by Marfey’s technique and an analysis of the acid hydrolysates using a chiral phase HPLC column. Among all the compounds, **6** and **8** showed medium cytotoxic activities against four tumor cell lines (SF-268, HepG-2, MCF-7, and A549), with IC_50_ values ranging from 31.03 ± 3.04 to 50.25 ± 0.54 µM. Meanwhile, they strongly inhibited α-glucosidase activities, with IC_50_ values of 35.73 ± 3.94 and 22.00 ± 2.45 µM, which were close to and even stronger than the positive control acarbose (IC_50_ = 32.92 ± 1.03 µM). Compounds **6**–**8** showed significant antibacterial activities against *Bacillus subtilis*, with MIC values of 10.26 ± 0.76 µM, 17.00 ± 1.25 µM, and 5.30 ± 0.29 µM, respectively. Compounds **9** and **12** exhibited potent radical scavenging activities against DPPH, with IC_50_ values of 12.23 ± 0.78 µM and 7.38 ± 1.16 µM. In addition, asperhiratide (**1**) was evaluated for anti-angiogenic activities in the in vivo zebrafish model, which showed a weak inhibitory effect on intersegmental vessel (ISV) formation.

## 1. Introduction

The coral reef ecosystem is one of the oldest ecological types with the highest species diversity and productivity on the Earth. It harbors abundant microbial communities, such as fungi, bacteria, actinomycetes, and cyanobacteria [1]. The complexity and extreme conditions marine organisms faced in their niche for survival provide them with a unique potential to produce a large range of compounds [2]. Among all coral microorganisms, fungi have emerged as an alternative source of promising bioactive agents, including alkaloids, terpenoids, polyketides, lipids, proteins, glycosides, and so on. At this point, more than 326 new structural metabolites have been reported from coral fungi [3,4].

Fungal species belonging to the genus *Aspergillus* have been widely studied. In the marine environment, several *Aspergillus* species have proved their potential to produce a plethora of secondary metabolites, which display a variety of pharmacological activities, such as antimicrobial, cytotoxic, anti-inflammatory, and antioxidant activity. From 2015 until December 2020, about 361 secondary metabolites were identified from different marine *Aspergillus* species [5]. However, it should be emphasized that there are still many rare and unstudied *Aspergillus* species, such as *Aspergillus hiratsukae*. In our continuing search for bioactive compounds from fungi collected in the South China Sea [6,7,8], we isolated the fungal strain *Aspergillus hiratsukae* SCSIO 5Bn_1_003 from an unidentified coral sample. Its crude extract exhibited various antibacterial activities in the pre-active screening, including *Bacillus subtilis*, *Staphylococcus aureus*, *Bacillus thuringiensis*, *Vibrio alginolyticus* XSBZ14, *Multi-resistant Pseudomonas aeruginosa*, and *Micrococcus lutea*. A further chemical investigation led to the isolation of three new metabolites, asperhiratide (**1**), asperhiratine (**2**), and asperhiratone (**3**) (Figure 1), as well as 10 known compounds. Herein, we report these compounds’ isolation, structure elucidation, and bioactivity evaluation.

## 2. Results

### 2.1. Structure Identification

Asperhiratide (**1**) was obtained as a white crystal, mp 269.4–271.8 °C. It was assigned a molecular formula of C_28_H_28_N_4_O_5_ on the basis of its positive HRESIMS data (*m*/*z* 501.2128 [M + H]^+^, calculated for 501.2132), indicating an index of hydrogen deficiency of 17. An analysis of the ^1^H and ^13^C NMR spectra (Table 1) revealed the presence of three amide *N*-H protons at *δ*_H_ 7.86 (1H, d, *J* = 9.9 Hz), 8.73 (1H, t, *J* = 5.7 Hz), and 9.79 (1H, s), one *N*-methyl at *δ*_H_ 2.56 (3H, s), three methylene groups, three phenyl rings, two methines, and four carbonyl carbons. An interpretation of the ^1^H, ^13^C, and 2D NMR data of **1** revealed the characteristics of a peptidyl structure. Through an analysis of the COSY and HMBC spectra of **1** (Figure 2), the aromatic protons at *δ*_H_ 7.51 (1H, dd, *J* = 7.6, 1.4 Hz), 7.12 (1H, td, *J* = 7.5, 1.1 Hz), 7.47 (1H, m), and 8.28 (1H, d, *J* = 8.2 Hz) were found to represent an *ortho*-disubstituted benzene ring. HMBC correlations from H-3 (*δ*_H_ 7.51) to C-1 (*δ*_C_ 170.1) and from *N*-H (*δ*_H_ 9.79) to C-7 (*δ*_C_ 136.9) led to the identification of an anthranilic acid (ABA) unit. Similarly, three other amino acid units, namely, *N*-Me-phenylalanine (*N*-Me-Phe), tyrosine (Tyr), and glycine (Gly), were completely assigned. Upon extensive analysis of these data, **1** was categorized as a cyclic tetrapeptide containing ABA, *N*-Me-Phe, Tyr, and Gly.

The amino acid sequence of **1** was deduced from HMBC correlations (Figure 2) between α-protons and neighboring residue *N*-H groups. The *N*-H proton (*δ*_H_ 9.79) of ABA showed an HMBC correlation with the carbonyl carbon of C-8, indicating that it was acylated by *N*-Me-Phe. The HMBC correlation from H_3–_17 to C-18 supported the connection of Tyr and *N*-Me-Phe. HMBC correlations from the *N*-H proton (*δ*_H_ 7.86) and H-19 of Tyr to C-27 supported the connection of the Tyr and Gly residues. The HMBC correlations from the *N*-H proton (*δ*_H_ 8.73) and H-28 of Gly to C-1 supported the connection of Gly and ABA residues. Finally, the cyclic planar structure of **1** was established as *cyclo*-[ABA-*N*-Me-Phe-Tyr-Gly].

The absolute configurations of the *N*-Me-Phe and Tyr residues were established by an HPLC-DAD and UPLC-MS analysis of the acid hydrolysate, derivatized with Marfey’s reagent (*N_α_*-(2,4-dinitro-5-fluorophenyl)-l-alaninamide, L-FDAA). The retention times of the acid hydrolysate of **1** derivatized with Marfey’s reagent on HPLC were 20.6, 23.2, 23.8, and 28.9 min, respectively. Additionally, a UPLC-MS analysis of the acid hydrolysate of **1** derivatized with Marfey’s reagent displayed retention times and positive ions at 24.2 min (*m*/*z* 434.1 [M + H]^+^, 867.3 [2M + H]^+^), and 29.3 min (*m*/*z* 432.2 [M + H]^+^, 863.3 [2M + H]^+^). Comparing these retention times with that of standards indicated the presence of l-*N*-Me, l-Tyr residues in **1**. Thus, the chemical structure of **1** was fully elucidated as *cyclo*-[ABA-l-*N*-Me-Phe-l-Tyr-Gly].

Delightedly, a single crystal of **1** suitable for X-ray diffraction analysis was obtained in the mixed solvent of CH_3_OH-acetone-H_2_O (3:2:1). The X-ray experiment using Cu Kα radiation (Figure 3, CCDC 2094392) with a good Flack parameter 0.03(8), unambiguously confirming the absolute stereochemistry to be l-*N*-Me-Phe and l-Tyr, which was consistent with the Marfey’s method result. Hence, the structure of **1** was assigned as *cyclo*-[ABA-l-*N*-Me-Phe-l-Tyr-Gly] and named asperhiratide.

Asperhiratine (**2**) was obtained as colorless crystal needles, mp 202.5–204.9 °C. It was assigned a molecular formula of C_35_H_46_O_9_ on the basis of its positive HRESIMS data (*m*/*z* 611.3227 [M + H]^+^, calculated for 611.3215), indicating an index of hydrogen deficiency of 13. The ^1^H-NMR spectra (Table 1) revealed the presence of four methyl singlets at *δ*_H_ 0.89 (3H, s), 0.97 (3H, s), 1.42(3H, s), and 1.27 (3H, d, *J* = 7.1 Hz), eight methylenes at *δ*_H_ 1.78 (1H, m), 1.42 (1H, d, *J* = 12.6 Hz), 2.10 (1H, m), 2.13 (1H, m), 1.80 (1H, m), 2.14 (1H, m), 1.68 (1H, m), 2.02 (1H, m), 1.67 (1H, m), 2.04 (1H, m), 1.82 (1H, m), 1.68 (1H, m), 1.75 (1H, m), 3.17 (1H, m), 4.25 (1H, dd, *J* = 9.0,7.7 Hz), and 3.89 (1H, t, *J* = 9.0 Hz), eight aliphatic methines at *δ*_H_ 3.84 (1H, m), 3.96 (1H, s), 2.12 (1H, m), 1.83 (1H, dd, *J* = 7.4, 3.2 Hz), 2.43 (1H, d, *J* = 9.2 Hz), 5.27 (1H, dd, *J* = 10.9, 2.1 Hz), 2.17 (1H, m), and 2.38 (1H,dd, *J* = 7.1, 2.7 Hz), and six aromatic protons at *δ*_H_ 5.82 (1H, t, *J* = 5.8 Hz), 8.10 (2H, dd, *J* = 8.3, 1.2 Hz), 7.52 (2H, t, *J* = 7.8 Hz), and 7.63 (1H, m). Its ^13^C NMR and DEPT data (Table 1) showed the presence of 35 carbon resonances, including four methyls, eight methylenes, and fourteen methines (six olefinic carbons), as well as nine quaternary carbon atoms (two olefinic carbons, three carbonyls, and four aliphatic carbons). Detailed analysis of its 1D and 2D NMR allowed for the assignment of three partial structures, including a typical ecdysteroid substructure with a *cis*-fused A/B ring junction and a 14*α*-hydroxy-7-ene-6-one skeleton [9], a *γ*-lactone side chain [10], and a monosubstituted benzoate group. All the above data taken together indicated that **2** was structurally related to the *γ*-lactone phytoecdysteroids 29-norcyasterone [11], with the only difference being the hydroxy substitution at C-22. C-22 (76.0 ppm) in 29-norcyasterone was chemically shifted downfield to 79.4 ppm in compound **2**. This indicated that the benzoate group was bounded to the C-22 hydroxy group. The conclusion was verified by the key HMBC correlation from H-22 (*δ*_H_ 5.27, dd, *J* = 10.9, 2.1 Hz) to C-1′ (*δ*_C_ 168.0). The NOE (Figure 2) correlations of H-5 (*δ*_H_ 2.40) with H_3–_19 (*δ*_H_ 0.97) and H_3_-18 (*δ*_H_ 0.89) and H-16a (*δ*_H_ 2.14) and H-22 (*δ*_H_ 5.27), indicated that they were located at the same side as the *α*-orientation. Furthermore, the NOE correlations of H-2 (*δ*_H_ 3.84) and H-9 (*δ*_H_ 1.83) with H-12b (*δ*_H_ 1.68) and H-17 (*δ*_H_ 2.43) and H-23b (*δ*_H_ 1.83) and H-27 (*δ*_H_ 2.38) led to their co-facial assignment as a *β*-orientation. Further confirmation of **2** was proven by single-crystal X-ray diffraction experimental data (Figure 3, CCDC 2004886), which were obtained from the mixed solvent of CH_3_OH-acetone (1:1). The X-ray experiment using Cu Kα radiation with a good Flack parameter [-0.11(18)] unambiguously confirmed the absolute stereochemistry to be 2*S*,3*R*,5*R*,9*S*,10*R*,13*R*,14*R*,17*S*,20*R*,22*R*,24*S*,27*S*. Hence, the structure of **2** was assigned as 2*S*,3*R*,5*R*,9*S*,10*R*,13*R*,14*R*,17*S*,20*R*,22*R*,24*S*,27*S*-29-norcyasterone-22-*O*-benzoate and named asperhiratine (Figure 2).

Asperhiratone (**3**) was obtained as a white powder. It was assigned a molecular formula of C_15_H_22_O_2_ on the basis of its positive HRESIMS data (*m*/*z* 235.1695 [M + H]^+^, calculated for 235.1693), indicating an index of hydrogen deficiency of 5. The ^1^H NMR spectrum (Table 1) showed two methyl singlets at *δ*_H_ 0.73 (3H, s) and 1.32 (3H, d, *J* = 7.1 Hz), six methylene singlets at *δ*_H_ 1.89 (1H, m), 2.29 (1H, m), 1.85 (1H, m), 2.59 (1H, m), 1.51 (1H, d, *J* = 10.5 Hz), 2.31 (1H, m), 1.68 (1H, d, *J* = 3.0 Hz), 1.92 (1H, m), 1.57 (1H, d, *J* = 3.0 Hz), 2.08 (1H, m), 4.58 (1H, s), and 4.70 (1H, s),and four methines at *δ*_H_ 2.56 (1H, m), 2.54 (1H, d, *J* = 5.5 Hz), 4.84 (1H, dd, *J* = 11.6, 3.2 Hz), and 2.43 (1H, m). The ^13^C NMR (Table 1) and DEPT spectra in combination with HMQC data revealed two methyl groups (*δ*_C_ 12.8, 17.6), six methylenes (*δ*_C_ 107.4, 28.9, 27.1, 24.1, 23.6, and 23.2), including one terminal ethylene, four methines (*δ*_C_ 84.9, 37.1, 47.8, and 36.5), including one oxymethine carbon, and three nonprotonated carbons (*δ*_C_ 175.1, 150.0, 47.9), including one lactone carbonyl and one quaternary olefinic. The aforementioned data occupied two degrees of unsaturation, and the remaining three degrees of unsaturation suggested the existence of three rings. Combined with these NMR signals, we could infer that **3** should be a tricyclic sesquiterpene lactone. In addition, the key HMBC correlations (Figure 2) between H_3_-15 and C-12/C-13/C-14, H-13 and C-11/C-12/C-15, and H_2_-12 and C-10/C-14/C-15, together with the COSY correlations of H-10 and H-11/H-12/H-13/H_3_-15, supported the presence of a methyl tetrahydro pyranone (ring C). Moreover, the spin systems of H_2_-4/H_2_-3/H-2/H_2_-7/H-6 in the COSY cross-peaks and the correlations from H-2 to C-4, C-6, and C-9, from H_2_-4 to C-2, C-6, and C-8, from H_2_-6 to C-2, C-4, C-7, C-9, and C-10, and from H_2_-7 to C-3, C-5, and C-1 in the HMBC spectrum, led to the construction of a six-membered ring A and a four-membered ring B with a terminal ethylene and a methyl group anchored to C-5 and C-1, respectively. Finally, the connection of ring B and ring C was confirmed by the HMBC correlations from H_2_-2, H_2_-6, and H_3_-9 to C-10, and from H-10 to C-1. Therefore, the planar structure of **3** was determined.

The relative configuration of **3** was determined by the NOE (Figure 2) correlation of H_2_-7 with H-10, which indicated their co-facial as *α*-orientations. Hitherto, the relative configuration of **3** was established as 1*R**,2*S**,6*S**. The calculated electronic circular dichroism (ECD) spectrum for (1*R*, 2*S*,6*S*,10*R*,13*S*)-**3** agreed well with that measured for **3** (Figure 4). Consequently, the absolute configuration of **3** could be deduced to be 1*R*,2*S*,6*S*,10*R*,13*S*. 

In addition to the isolation of the above new compounds **1**–**3**, 10 known compounds, including schizaeasterone A (**4**) [12], terretonin (**5**) [13], demethylincisterol A_2_ (**6**) [14], asperophiobolin E (**7**) [15], butyrolactone I (**8**) [16], butyrolactones VI (**9**) [17], *epi*-aszonalenins B (**10**) [18], flavoglaucin (**11**) [19], 6,8-dimethoxy-3-methylisocoumar (**12**) [20], and methyl shikimate (**13**) [21] were also isolated and identified from this fungus. Their structures (Figure 1) were elucidated through a comparison of their NMR and MS data with reported literature.

### 2.2. Bioactivity Evaluation

All compounds were evaluated for their in vitro cytotoxicity against four human cancer lines, SF-268 (human glioblastoma carcinoma), MCF-7 (breast cancer), HepG-2 (liver cancer), and A549 (lung cancer), as well as antioxidative activity, α-glucosidase inhibitory activity, and antimicrobial activity against the bacteria *Staphylococcus aureus* and *Bacillus subtilis* (Appendix A). Compounds **6** and **8** showed moderate cytotoxic activities against all four tumor cell lines, with IC_50_ values ranging from 31.03 ± 3.04 to 50.25 ± 0.54 µM (Table 2). Meanwhile, they exhibited significant inhibitory activity against *α*-glucosidase, with IC_50_ values of 35.73 ± 3.94 and 22.00 ± 2.45 µM, which are even better or comparable to that of the positive control acarbose (IC_50_ = 32.92 ± 1.03 µM) (Table 3). Compounds **6**–**8** showed significant antibacterial activities against *Bacillus subtilis*, with MIC values from 5.30 ± 0.29 to 10.26 ± 0.76 µM (Table 4). Compounds **7** and **8** showed weak activity against *Staphylococcus aureus*, with MIC values of 102.86 ± 4.50 and 59.54 ± 0.50 µM, respectively. Compounds **9** and **12** showed significant DPPH radical scavenging activity, with EC_50_ values ranging from 12.23 ± 0.78 to 7.38 ± 1.16 µM (Table 3). In addition, Compound **1** was also evaluated for anti-angiogenic activities in the in vivo zebrafish model, which showed a weak inhibitory effect on intersegmental vessel (ISV) formation (Figure 5).

## 3. Materials and Methods

### 3.1. General

NMR spectra were recorded on an AVANCE III HD-700 spectrometer (Bruker, Billerica, MA, USA) with TMS as the internal standard. HR-ESI-MS spectra data were gathered on a MaXis quadrupole time-of-flight mass spectrometer (Bruker Biospin GmbH, Rheinstetten, Germany). Crystallographic data were collected on a Rigaku XtaLAB AFC12 single-crystal diffractometer (Rigaku, Kyoto, Japan) using Cu Kα radiation. Optical rotations were measured with an MCP500 automatic polarimeter (AntonPaar, Graz, Austria) with MeCN as the solvent. UV spectra were recorded on a UV-2600 spectrometer (Shimadzu, Tokyo, Japan). Circular dichroism spectra were measured with a Chirascan circular dichroism spectrometer (Applied Photophysics, Ltd., Surrey, Leatherhead, UK) with the same concentration of UV measurement (Pathlength 10 mm, Applied Photophysics, Ltd., Surrey, Leatherhead, UK). HPLC purification was performed on an Agilent 1260 HPLC equipped with a DAD detector, using an ODS column (YMC-pack ODS-A, 10 × 250 mm, 5 µM, 3 mL/min). Silica gel (200–300 and 300–400 mesh) and Sephadex LH-20 for column chromatography (CC) were purchased from Qingdao Marine Chemical Group Co. (Qingdao, China) and GE Healthcare (Uppsala, Sweden), respectively. All solvents used in CC and HPLC were of analytical grade (Tianjin Damao Chemical Plant, Tianjin, China) and chromatographic grade (MERDA, Beijing, China), respectively.

### 3.2. Fungal Materials

The strain *Aspergillus hiratsukae* SCSIO 5Bn_1_003 was isolated from a coral sample collected from the South China Sea in September 2019. The internal transcribed spacer (ITS1-5.8S-ITS2) regions of the strain were amplified with the universal ITS primers, ITS1 (5′-CTTGGTCATTTAGAGGAAGTAA-3′) and ITS4 (5′-TCCTCCGCTTATTGATATGC-3′), using polymerase chain reaction (PCR). The amplified products were submitted for sequencing and aligned with the sequences in the GenBank by Basic Local Alignment Search Tool (BLAST) programs to find out the sequence homology with closely related organisms. When the top three matching BLAST hits were from the same species and were ≥98% similar to the query sequence, this species name was assigned to the selected isolate. Finally, the strain was identified as *A. hiratsukae* under the accession number KY806121.1. The working strain was deposited with the Research Network for Applied Microbiology (RNAM) Center for Marine Microbiology, South China Sea Institute of Oceanology, Chinese Academy of Sciences.

### 3.3. Fermentation, Extraction, and Purification

The strain was reactivated with a PDA plate, transferred to a 500 mL flask containing 150 mL of fermentation medium (potato extract 6.0 g, glucose 20.0 g, artificial sea salt 30.0 g, distilled water 1.0 L), and cultured at 28 °C, 180 r/min shaker, for 3 days to obtain a seed solution. Then, 5 mL of the seed solution was inoculated into 1000 mL flasks containing 300 mL of fermentation medium, and cultured at 28 °C static for 30 days to obtain 50 L of fermentation broth. The fermentation broth was filtered through a sieve and divided into the filtrate and mycelia. The filtrate was concentrated under reduced pressure, and was extracted with an equal volume of ethyl acetate three times to obtain an ethyl acetate extract of the fermentation broth. The mycelia were extracted with an 80% aqueous acetone solution three times, concentrated under reduced pressure until acetone-free, and extracted with an equal volume of ethyl acetate three times to obtain an ethyl acetate extract of mycelia. The ethyl acetate extracts were combined and concentrated to dryness under reduced pressure to give 25.8 g of crude product.

The crude extract was subjected to silica gel CC eluted with petroleum ether/ethyl acetate in linear gradient (40:1–0:1) to obtain eight fractions (Frs.1–8) on the basis of TLC profiles. Fr.2 (0.7 g) was subjected to semi-HPLC (85% CH_3_OH/H_2_O) to obtain three subfractions (Frs.2.1–2.3). Fr.2.1 was further purified via HPLC (75% CH_3_OH/H_2_O) to yield **3** (0.8 mg). Fr.3 (7.6 g) was separated by silica gel CC with CH_2_Cl_2_/MeOH (1:0 to 0:1) to obtain ten subfractions (Frs.3.1–3.10). Then, Fr.3.1 (162 mg) was further purified with HPLC (70% CH_3_CN/H_2_O) to yield **2** (2.2 mg) and **10** (2.2 mg). Fr.3.2 (4.5 g) was subjected to the Sephadex LH-20 (CH_2_Cl_2_/MeOH 1:1) to obtain six subfractions (Frs.3.2.1–3.2.6). Fr 3.2.3 was identified as **8** (3.7 g). Fr 3.2.1 was further purified by HPLC (50% CH_3_OH/H_2_O) to yield **7** (5.6 mg), **9** (5.6 mg), **12** (1.8 mg), and **13** (3.6 mg). Fr 3.2.4 was further purified with HPLC (95% CH_3_OH/H_2_O) to yield **11** (3.2 mg). Fr.4 was subjected to the Sephadex LH-20 (CH_2_Cl_2_/MeOH 1:1) to obtain five subfractions (Frs.4.1–4.5). Then, Fr.4.3 (26.5 mg) was further purified with HPLC (65% CH_3_CN/H_2_O) to yield **4** (6.3 mg). Fr.4.5 (322 mg) was purified via repeated HPLC (38% CH_3_CN/H_2_O) to yield **1** (180.1 mg), **5** (2.8 mg), and **6** (2.4 mg).

### 3.4. Spectral Data

Asperhiratide (**1**): white crystal. ${\left[\alpha \right]}_{D}^{25}$ = −10.2 (c 2.0, MeOH); UV (MeOH) λ_max_ (log *ε*) 204.20 (1.84), 208.80 (1.81), 211.80 (1.81), 285.60 (0.16), 387.40 (−0.01) nm; HRESIMS at *m*/*z* 501.2128 [M + H]^+^, calculated for C_28_H_29_N_4_O_5_, 501.2132. For ^1^H and ^13^C NMR, see Table 1.

Asperhiratine (**2**): colorless crystal needles. ${\left[\alpha \right]}_{D}^{25}$ = +26.3 (c 1.0, MeOH); UV (MeOH) λ_max_ (log *ε*) 201.60 (1.39), 232.00 (1.58), 315.40 (0.03), 326.80 (0.02), 374.6 (0.01) nm; HRESIMS at *m*/*z* 611.3227 [M + H]^+^, calculated for C_35_H_47_O_9_, 611.3215. For ^1^H and ^13^C NMR, see Table 1.

Asperhiratone (**3**): white powder. ${\left[\alpha \right]}_{D}^{25}$ = −1.30 (c 1.0, MeOH); UV (MeOH) λ_max_ (log *ε*) 203.80 (1.63) nm; HRESIMS at *m*/*z* 235.1695 [M + H]^+^, calculated for C_15_H_23_O_2_, 235.1693. For ^1^H and ^13^C NMR, see Table 1.

### 3.5. Preparation and Analysis of Marfey Derivatives

Asperhiratide (**1**) (2.0 mg) was hydrolyzed via heating in HCl (6 M; 1 mL) at 110 °C for 24 h. After cooling, the solution was evaporated to dryness and redissolved in H_2_O (50 µL). To the acid hydrolysate solution (or to 50 µL of a 50 mM solution of the respective amino acid) was added a 1% (*w/v*) solution (100 µL) of FDAA (Marfey’s reagent, NR-(2,4-dinitro-5-fluorophenyl)- l-alaninamide) in acetone. After the addition of the NaHCO_3_ solution (1 M; 20 µL), the mixture was incubated for 1.0 H at 40 °C. The reaction was stopped by the addition of HCl (2 M; 10 µL), the solvents were evaporated to dryness, and the residue was redissolved in MeOH-H_2_O (1:1; 1 mL). An aliquot of this solution (10 µL) was analyzed by HPLC (YMC-pack ODS-A, 250 × 4.6, 5 µM; solvents: (A) water + 1% TFA, (B) MeCN; linear gradient: 0 min 5% B, 25 min 48% B, 50 min 65% B; 25 °C; 1 mL min^−1^; 25 °C). The retention times (min) of the amino acid derivatives were as follows: l-Tyr (24.2 min), d-Tyr (26.3 min), and *N*-Me-l-Phe (29.3 min).

### 3.6. X-ray Crystal Structure Analysis

Crystallographic data for **1** and **2** were collected on a Rigaku XtaLAB AFC12 single-crystal diffractometer using Cu Kα radiation. The crystal was kept at 99.9(5) K during data collection. Using Olex2, the structure was solved with the ShelXT structure solution program using intrinsic phasing, and was refined with the ShelXL refinement package using least-squares minimization. Crystallographic data for **1** and **2** have been deposited in the Cambridge Crystallographic Data Centre database (deposition numbers 2,094,392 and 2004886). Copies of the data can be obtained free of charge from the CCDC at www.ccdc.cam.ac.uk (accessed on 9 July 2021).

Crystal data for **1**: C_28_H_28_N_4_O_5_, *M* = 500.54, orthorhombic, space group P212121 (no.19), *a* = 8.18580(10) Å, *b* = 11.06650(10) Å, *c* = 27.4876(3) Å, *V* = 2490.05(5) Å3, *Z* = 4, *T* = 100.00(10) K, *μ* (Cu Kα) = 0.763 mm^−1^, *D*
_calc_ = 1.335 g/cm^3^, 13,956 reflections measured (6.432° ≤ 2Θ ≤ 149.084°), 4882 unique (*R*
_int_ = 0.0317, *R*
_sigma_ = 0.0311) which were used in all calculations. The final *R*_1_ was 0.0343 (*I* > 2σ*(I)*) and *wR*_2_ was 0.0887 (all data). The goodness of fit on *F*^2^ was 1.076. Flack parameter = 0.03(8), melting point: 269.4–271.8°.

Crystal data for **2**: C_35_H_46_O_9_, *M* =699.76, monoclinic, space group P21 (no.4), *a* = 6.3701(3) Å, *b* = 12.9223(8)Å, *c* = 21.9253(9)Å, β = 92.335(4)°, *V* = 1803.31(16) Å3, *Z* = 2, *T* = 100.0(10) K, *μ*(Cu Kα) = 0.825 mm^−1^, *D*
_calc_ = 1.289 g/cm^3^, 17,416 reflections measured (7.944° ≤ 2Θ ≤ 148.904°), 7011 unique (*R* _int_ = 0.0880, *R* _sigma_ = 0.0746) which were used in all calculations. The final *R*_1_ was 0.0675 (*I* > 2σ(I)) and *wR*_2_ was 0.1981 (all data). The goodness of fit on *F*^2^ was 1.074. Flack parameter = −0.11(18), melting point: 202.5–204.9°.

### 3.7. Bioassays

#### 3.7.1. Cytotoxic and Antioxidative Assays

The details of the experimental procedures for cytotoxic and antioxidative bioassays were described previously in our former paper [22].

#### 3.7.2. Antibacterial Assay

All the compounds were tested for antibacterial activity against *Staphylococcus aureus* and *Bacillus subtilis* using the Mueller–Hinton broth microdilution method [23]. Ciprofloxacin was used as a positive control.

#### 3.7.3. α-Glucosidase Inhibitory Assay

The anti-α-glucosidase activity of the active compounds was evaluated by measuring the increment of absorbance due to the hydrolysis of PNP-G by *α*-glucosidase at the wavelength of 410 nm with a microplate reader, according to the referenced described method [24].

#### 3.7.4. Zebrafish Assays for In Vivo Anti-Angiogenic Activity

FLI-1: EGFP transgenic zebrafish larvae, purchased from the Molecular Toxicology Research Center of Oregon State University (Covallis, OR, USA), exhibited green fluorescence under fluorescence spectroscopy (Fluorescence Inverted Microscope, Nikon TE2000-U, Tokyo, Japan). The maintenance of zebrafish and the collection of embryos followed the method in the literature [25]. One day after fertilization, the embryos were dechorionated with tweezers in a Petri dish coated with 1% agarose and distributed into small Petri dishes (35 × 10 mm) with 3 mL of assay solution, depending on the drug treatment. The embryos receiving a mixture of DMSO (0.1%) were used as the vehicle-treated control and were equivalent to a “no treatment” group. All the experiments were repeated at least three times, with 20 embryos per group.

## 4. Conclusions

In summary, the chemical investigation of a soft coral-derived fungus *Aspergillus hiratsukae* SCSIO 5Bn_1_003, collected from the South China Sea, led to the discovery of 13 compounds with diverse chemical structures, comprising cyclic peptide, ecdysteroid, sesquiterpene lactone, triterpene, butyrolactones, benzaldehyde, bisphthalate, and coumarin. The results demonstrated the enormous biosynthetic potential of this fungus strain. Asperhiratide (**1**) contains a rare Gly residue and showed a weak inhibitory effect on intersegmental vessel (ISV) formation. Asperhiratine (**2**) is a typical phytoecdysteroid which possesses a cholest-7-en-6-one carbon skeleton, a *γ*-lactone, and a benzoate group. Most phytoecdysteroids are present in a large number of plant families within the angiosperms, gymnosperms, ferns, and terrestrial fungi, but are very rare in marine fungi. Asperhiratone (**3**) represents an unprecedented sesquiterpene lactone, which possesses a new 6/4-fused carbocyclic core skeleton containing a lactone ring system. In addition, compounds **6**–**9** and **12** exhibited various activities, such as antimicrobial, cytotoxic, *α*-glucosidase inhibitory, and antioxidant activity. The above findings demonstrated the huge potentials of coral-associated fungi for the discovery of structurally novel and pharmacologically active natural products.

## Figures and Tables

**Figure 1 marinedrugs-20-00150-f001:**
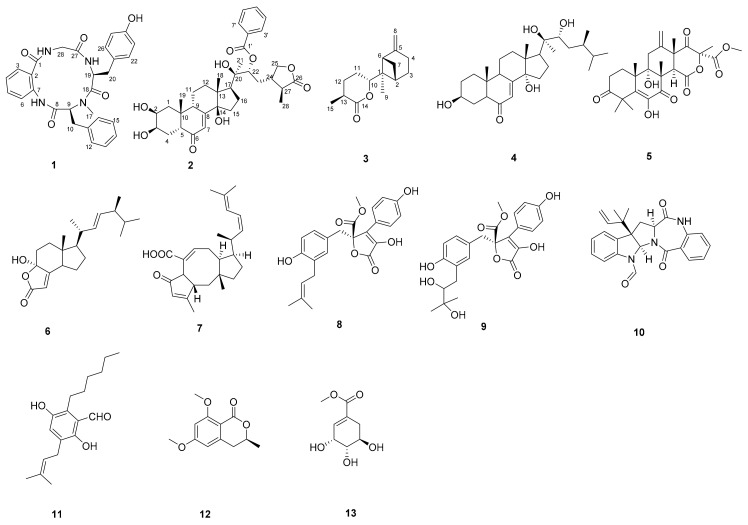
Chemical structures of compounds **1**–**13**.

**Figure 2 marinedrugs-20-00150-f002:**
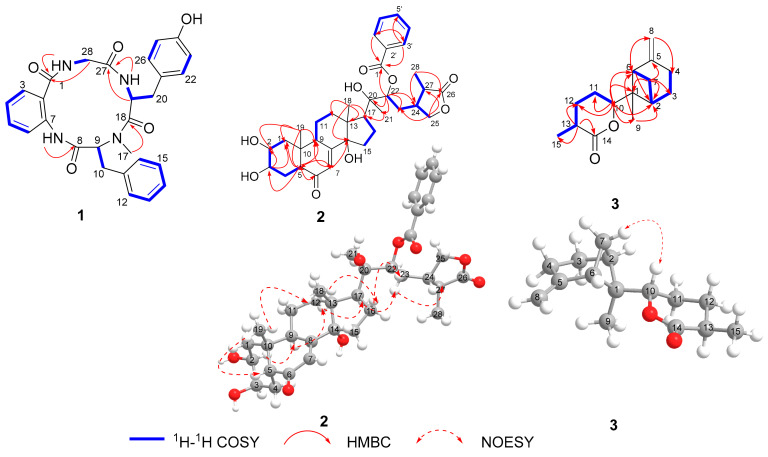
Key ^1^H-^1^H COSY, HMBC, and NOESY correlations of compounds **1**–**3**.

**Figure 3 marinedrugs-20-00150-f003:**
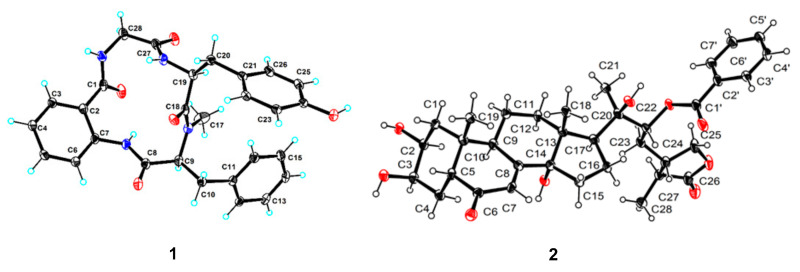
X-ray crystallographic structures of compounds **1**–**2**.

**Figure 4 marinedrugs-20-00150-f004:**
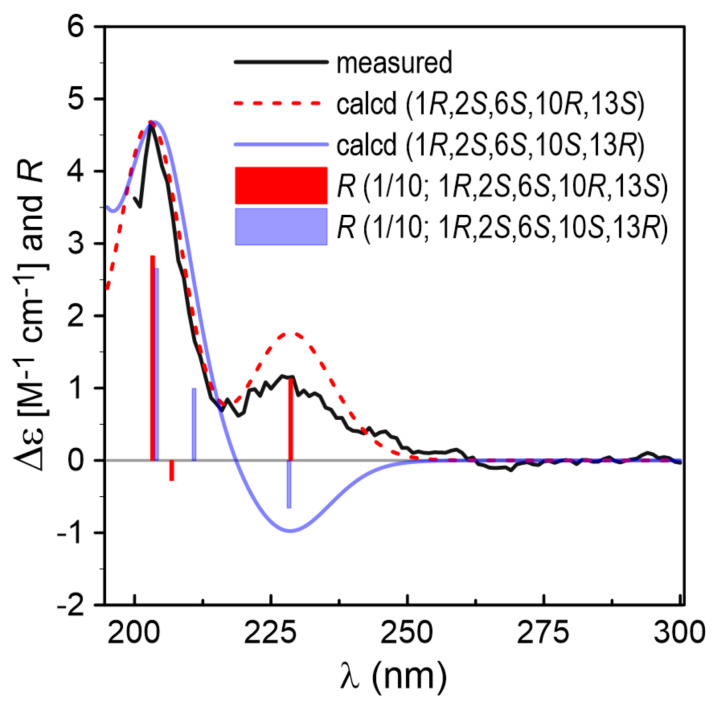
Comparison of the experimental ECD spectrum of **3** with the calculated spectra of (1*R*,2*S*,6*S*,10*R*,13*S*)- and (1*R*,2*S*,6*S*,10*S*,13*R*)-**3**. *R* represents computed rotatory strengths.

**Figure 5 marinedrugs-20-00150-f005:**
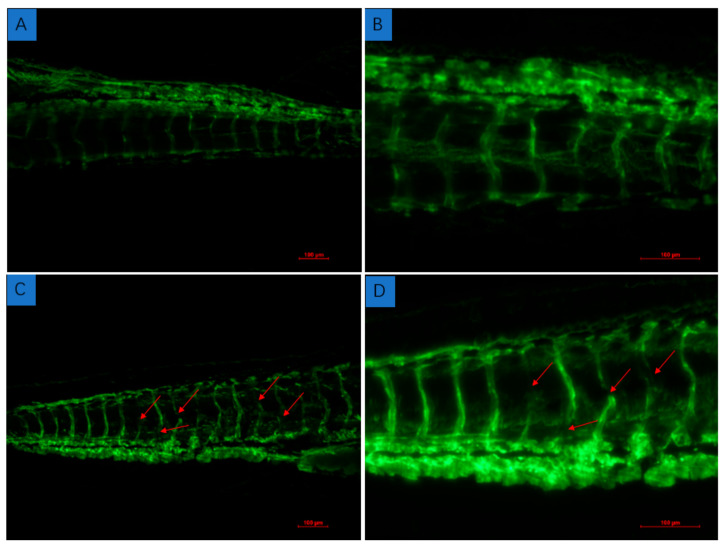
Inhibitory effect of compound **1** on embryonic vascular development of FLI-eGFP transgenic zebrafish. Picture (**A**,**B**) (partial enlargement) are positive controls; picture (**C**,**D**) (partial enlargement) are treated by compound **1**. The red arrows show the damage to vascular development.

**Table 1 marinedrugs-20-00150-t001:** The ^1^H and ^13^C NMR data for **1**–**3** (700 MHz for ^1^H and 175 MHz for ^13^C NMR in CDCl_3_).

No.	1		2		3	
	*δ* _C_	*δ*_H_, Mult. (*J*, Hz)	*δ* _C_	*δ* _H_	*δ* _C_	*δ* _H_
1	170.1	-	37.3	1.80 m 1.45 d (12.6)	47.9	-
2	124.4	-	68.7	3.84 m	37.1	2.56 m
3	126.6	7.51 dd (7.6, 1.4)	68.5	3.96 s	23.2	1.85 m 2.59 m
4	122.4	7.12 td (7.5,1.1)	34.2	2.10 m 2.13 m	23.6	1.89 m 2.29 m
5	131.2	7.47 m	51.8	2.40 m	150.0	-
6	119.9	8.28 d (8.2)	206.3	-	47.8	2.54 d (5.5)
7	136.9	-	122.3	5.83 d (2.4)	27.1	1.51 d (5.5) 2.31 m
8	167.8	-	167.5		107.4	4.58 s 4.70 s
9	68.3	4.01 dd (11.1, 3.6)	35.1	1.83 dd (7.4, 3.2)	12.8	0.73 s
10	32.6	2.86 dd (13.2, 11.3) 3.1 dd (13.5, 3.4)	39.3	-	84.9	4.84 dd (11.6, 3.2)
11	138.8		21.5	1.80 m 2.14 m	24.1	1.68 d (3.0), 1.92 m
12	129.1	6.57 d (7.0)	31.7	1.68 m 2.03 m	28.9	1.57 d (3.0) 2.08 m
13	128.2	7.07 dd (11.2, 4.3)	50.9		36.5	2.43 m
14	125.9	7.10 dt (2.5, 2.0)	85.3		175.1	-
15	128.2	7.07 dd (11.2, 4.3)	32.6	1.67 m 2.16 m	17.6	1.32 d (7.1)
16	129.1	6.57 d (7.0)	21.6	2.14 m 1.68 m		
17	39.0	2.56 s	50.9	2.43 d (9.2)		
18	168.2		18.0	0.89 s		
19	49.2	5.01 td (10.1, 5.3)	24.4	0.97 s		
20	36.7	2.92 dd (13.1,10.5)	77.7			
21	127.2		21.7	1.42 s		
22	130.4	7.01, d (8.4)	79.4	5.27 dd (10.9, 2.1)		
23	115.3	6.73 d (8.4)	34.1	2.11 m 1.83 m		
24	156.2	-	43.2	2.20 m		
25	115.3	6.73 d (8.4)	73.5	4.25 dd (9.0, 7.7) 3.89 t (9.0)		
26	130.4	7.01 d (8.4)	181.8	-		
27	170.2	-	41.6	2.38 dd (7.1, 2.7)		
28	45.5	3.65 dd (5.5, 3.2)	14.5	1.27 d (7.1)		
1′			168.0			
2′			131.7			
3′			130.8	8.10 dd (8.3,1.2)		
4′			129.7	7.51 dd (11.0, 4.6)		
5′			134.4	7.63 m		
6′			129.7	7.51 dd (11.0, 4.6)		
7′			130.8	8.10 dd (8.3,1.2)		

**Table 2 marinedrugs-20-00150-t002:** Cytotoxic activity of compounds against tumor cells ^a^.

Compounds	IC_50_ (μM)			
	SF-268	MCF-7	HepG-2	A549
**6**	34.75 ± 0.95	39.40 ± 0.52	43.65 ± 2.08	50.25 ± 0.54
**8**	31.41 ± 1.98	38.57 ± 2.48	31.03 ± 3.04	32.31 ± 2.36
Adriamycin	1.19 ± 0.03	2.02 ± 0.04	1.99 ± 0.07	1.73 ± 0.04

^a^ The results are mean ±SD (SD = standard deviation). Positive control: adriamycin.

**Table 3 marinedrugs-20-00150-t003:** Antioxidative and *α*-glucosidase inhibitory activities of compounds.

Compounds	Antioxidative Activity EC_50_ (µM)	*α*-Glucosidase Inhibitory IC_50_ (µM)
**6**	-	35.73 ± 3.94
**8**	-	22.00 ± 2.45
**9**	12.23 ± 0.78	-
**12**	7.38 ± 1.16	-
Ascorbic acid ^a^	11.35 ± 0.56	-
Acarbose ^b^	-	32.92 ± 1.03

^a^ Positive control for antioxidative activity; ^b^ positive control for α-glucosidase inhibitory activity.

**Table 4 marinedrugs-20-00150-t004:** The antibacterial activities of compounds ^a^.

Bacterial Species	MIC (µg·mL^−1^)			
	6	7	8	Ciprofloxacin
*Staphylococcus aureus*	-	102.86 ± 4.50	59.54 ± 0.50	0.07 ± 0.001
*Bacillus subtilis*	10.26 ± 0.76	17.00 ± 1.25	5.30 ± 0.29	0.09 ± 0.003

^a^ The results are mean ± SD (SD = standard deviation). Positive control: ciprofloxacin.

## Data Availability

Data are contained within the article or Supplementary Material.

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
