# Peer review of "Diverse Secondary Metabolites from the Coral-Derived Fungus Aspergillus hiratsukae SCSIO 5Bn1003"

_marinedrugs, 2022, doi:10.3390/md20020150_

Round 1

Reviewer 1 Report

This manuscript presents the structure elucidation and biological activity of new metabolites isolated from the soft coral-derived fungus, including X-ray analysis. This manuscript needs some modifications as follows.

(1) Lines 69 and 274: The HRESIMS data of compound 1differ between Results (m/z 501.2128 [M+H] +, calcd for 501.2123) and Materials and Methods (m/z 501.2132 [M+H] +, calcd for C28H28N4O5, 501.2128). Please accurately describe the MS data for compound 1.

(2) Line 189: The authors stated that all compounds were evaluated for cytotoxicity, antioxidative activity, a-glucosidase inhibitory, and antimicrobial activity. Please list the bioactivity of all compounds in Tables.

(3) Line 275: ‘C28H28N4O5’ needs to be modified to ‘C28H29N4O5’.

(4) Line 278: ‘C35H46O9’ needs to be modified to ‘C35H47O9’.

(5) Supplementary Materials: MS spectra for compounds 2-3 should be provided in full (m/z 100-1000 and above).

Author Response

Dear Reviewer,

Thank you for your help with our manuscript entitled " Diverse Secondary Metabolites from the Coral-Derived Fungus Aspergillus hiratsukae SCSIO 5Bn1003". We have revised the manuscript with the "Track Changes" function in Microsoft Word. According to these comments, our explanations for the corrections are listed as follows.

Point 1: Lines 69 and 274: The HRESIMS data of compound 1 differ between Results (m/z 501.2128 [M+H] +, calcd for 501.2123) and Materials and Methods (m/z 501.2132 [M+H] +, calcd for C28H28N4O5, 501.2128). Please accurately describe the MS data for compound 1.

Response 1: We carefully checked the HRESIMS data of 1 and confirmed it was m/z 501.2128 [M+H] +, calcd for 501.2132, and the corresponding errors have been corrected in Lines 67-68 and 275.

Point 2: Line 189: The authors stated that all compounds were evaluated for cytotoxicity, antioxidative activity, α-glucosidase inhibitory, and antimicrobial activity. Please list the bioactivity of all compounds in Tables.

Response 2: We have listed the primary screening bioactivity data of all compounds in Tables S1-S3 in Supplementary Materials (page 20-21). Due to the poor activity of many compounds in the primary screening, they are not re-screened for second time and not calculated the IC50, EC50, MIC, etc. Therefore, only meaningful bioactivity data (compounds 6-9 and 12) were listed in the manuscript, other inactive-compounds’ bioactivity data provided in Supplementary Materials.

Point 3: Line 275: ‘C28H28N4O5’ needs to be modified to ‘C28H29N4O5’.

Response 3: We have modified“C28H28N4O5” to “C28H29N4O5” in Line 275.

Point 4: Line 278: ‘C35H46O9’ needs to be modified to ‘C35H47O9’.

Response 4: We have modified “C35H46O9” to “C35H47O9” in Line 278.

Point 5: Supplementary Materials: MS spectra for compounds 2-3 should be provided in full (m/z 100-1000 and above)

Response 5: We have provided in full (m/z 100-1000 and above) for compounds 2-3 in Supplementary Materials (Figure S15 in page 11 and Figure S23 in page 15).

Some other grammatical and expressive errors also have been revised. Thank you and all the reviewers for the kind suggestion.

Best regards

Dr. Fazuo Wang (Prof.) and Si Zhang (Prof.)

South China Sea Institute of Oceanology

Chinese Academy of Sciences

164 West Xingang Road, Guangzhou, China

Tel : +86-20-3406-3746

+86-20-8902-3103

E-mail:[email protected]

[email protected]

Reviewer 2 Report

The manuscript entitled "Diverse Secondary Metabolites from the Coral-Derived Fungus 2 Aspergillus hiratsukae SCSIO 5Bn1003" regards the isolation of three new compounds and ten known compounds and the assignment of their structure as well as the absolute configurations of the chiral centers of the new ones. The isolation of new compounds is always important and the determination of their biological activities could lead to the discovery of potential new drugs. The manuscript is clear and well written although English style should be revised since there are grammatical errors in some sentences. The experimental part is complete although the authors should add in the general part of materials and methods the type of instrument used for determining the optical rotatory power.  The quality of spectra reported in the supplementary material is good and purified compounds are sufficiently pure. Typographical errors are also present as for example compound numbers are not always in bold style especially in the abstract. Also the scientific names of species should be reported in italic.

Author Response

Dear Reviewer,

Thank you for your help with our manuscript entitled " Diverse Secondary Metabolites from the Coral-Derived Fungus Aspergillus hiratsukae SCSIO 5Bn1003". We have revised the manuscript with the "Track Changes" function in Microsoft Word. According to these comments, our explanations for the corrections are listed as follows.

Point 1: English style should be revised since there are grammatical errors in some sentences.

Response 1: We have carefully checked the whole manuscript and corrected the English writing and grammar errors, mainly see the Abstract and Introduction for details.

Point 2: The authors should add in the general part of materials and methods the type of instrument used for determining the optical rotatory power.

Response 2: We have added the type of instrument used for determining the optical rotatory power in the general part of materials and methods in Line 219-220.

Point 3: Typographical errors are also present as for example compound numbers are not always in bold style especially in the abstract. Also, the scientific names of species should be reported in italic..

Response 3: The typographical mistakes in the manuscript have been carefully checked and corrected, such as bold style for compounds number and italic style for the scientific names of species. Actually, they were originally correct in the first version we submitted to the system, however in the version you received for revisions, it may be displayed font errors due to some unkonwn system reasons.

Some other grammatical and expressive errors also have been revised. Thank you and all the reviewers for the kind suggestion.

Best regards

Dr. Fazuo Wang (Prof.) and Si Zhang (Prof.)

South China Sea Institute of Oceanology

Chinese Academy of Sciences

164 West Xingang Road, Guangzhou, China

Tel : +86-20-3406-3746

+86-20-8902-3103

E-mail:[email protected]

[email protected]
